# Multifunctional Analysis of Chia Seed (*Salvia hispanica* L.) Bioactive Peptides Using Peptidomics and Molecular Dynamics Simulations Approaches

**DOI:** 10.3390/ijms23137288

**Published:** 2022-06-30

**Authors:** José E. Aguilar-Toalá, Abraham Vidal-Limon, Andrea M. Liceaga

**Affiliations:** 1Departamento de Ciencias de la Alimentación, División de Ciencias Biológicas y de la Salud, Universidad Autónoma Metropolitana, Unidad Lerma. Av. de las Garzas 10. Col. El Panteón, Lerma de Villada 52005, Estado de México, Mexico; j.aguilar@correo.ler.uam.mx; 2Red de Estudios Moleculares Avanzados, Instituto de Ecología A.C. (INECOL), Carretera Antigua a Coatepec 351, El Haya, Xalapa 91073, Veracruz, Mexico; abraham.vidal@inecol.mx; 3Protein Chemistry and Bioactive Peptides Laboratory, Department of Food Science, Purdue University, 745 Agriculture Mall Dr., West Lafayette, IN 47907, USA

**Keywords:** bioactive peptides, ensemble docking, multifunctional bioactivities, molecular dynamics simulations, chronic diseases

## Abstract

Chia seed peptides (CSP) can be a source of multifunctional biopeptides to treat non-communicable diseases. However, interactions and binding affinity involved in targeting specific receptors remains unexplored. In this study, molecular simulation techniques were used as virtual screening of CSP to determine drug-like candidates using a multi-target-directed ligand approach. CSP fraction with the best bioactivities in vitro was sequenced. Then, a prediction model was built using physicochemical descriptors (hydrophobicity, hydrophilicity, intestinal stability, antiangiogenic, antihypertensive, and anti-inflammatory) to calculate potential scores and rank possible biopeptides. Furthermore, molecular dynamics simulations (MDS) and ensemble molecular docking analysis were carried out using four human protein targets (ACE, angiotensin converting enzyme; VEGF, vascular endothelial growth factor; GLUC, glucocorticoid and MINC, mineralocorticoid receptors). Five known-sequence peptides (NNVFYPF, FNIVFPG, SRPWPIDY, QLQRWFR, GSRFDWTR) and five de novo peptides (DFKF, DLRF, FKAF, FRSF, QFRF) had the lowest energy score and higher affinity for ACE and VEGF. The therapeutic effects of these selected peptides can be related to the inhibition of the enzymes involved in angiogenesis and hypertension, due to formation of stable complexes with VEGF and ACE binding sites, respectively. The application of MDS is a good resource for identifying bioactive peptides for future experimental validation.

## 1. Introduction

In recent years, there has been an increased interest in finding bioactive peptides that can prevent the risk of chronic diseases and/or boost the immune system. The advancement of peptidomics and bioinformatics in food science has enabled integrated studies to be carried out for the rapid development of food-derived bioactive peptides [1]. Moreover, the application of an integrated approach in the study of bioactive peptides allows for their production to be optimized, leads to the discovery of peptides of specific interest, and contributes to the understanding of the mechanism of action between bioactive peptides and human target receptors [2]. Accordingly, this integrated approach is more cost-effective and time-saving compared with the standard approach used, which involves more laborious experimental tests to study bioactive peptides [2,3]. In view of the above, these advancements have provided important tools for efficient discovery of novel peptides with profitable biological activities, for the analysis of peptide datasets, and an understanding of their structure-activity relationships [3]. The latter will allow researchers to explore the complete picture and/or entire spectrum of bioactivities from any protein source.

Chia (*Salvia hispanica* L.) seeds are recognized as an emerging protein source. Aside from its role as dietary nutrients, it is being progressively recognized for its bioactive properties due to the presence of essential and non-essential amino acids, ω-3 fatty acids, phytosterol, and dietary fiber [4,5,6]. Recently, our research group demonstrated that chia seed peptides, obtained by sequential and microwave-assisted proteolysis, exhibited prominent chemical and cellular antioxidant [7], antimicrobial [8], and anti-aging [9] properties. Similarly, other studies have reported that chia seed peptides possess in vitro antioxidant [10], antibacterial [11], and anti-inflammatory [12] activities. Therefore, our aim was to apply bioinformatics and in silico methodologies (i.e., physicochemical properties prediction, molecular dynamics and molecular docking simulations) to predict potential bioactivities associated with chia seed peptides. The outcomes of this study will generate new knowledge into their mechanism of action associated with intermolecular interactions within enzymes involved in well-recognized chronic diseases such as hypertension and inflammation. 

## 2. Results and Discussion

Chia seed is an interesting protein source that can be studied to identify peptides and explore their potential bioactivities. Since chia seed peptides (CSP) obtained by sequential and microwave-assisted proteolysis have demonstrated, in our laboratory, to have good chemical and cellular antioxidant [7], antimicrobial [8], and anti-aging [9] in vitro activities, they were selected for peptidomics sequencing and bioinformatic analysis. Many traditional in silico methods are challenging to adapt to certain classes of biomolecules, mostly when the biomolecules consist of large and highly flexible chemical moieties (like peptides, toxins, and antibodies). For example, generating libraries of large peptides is currently feasible, due to massive mass spectrometry techniques and/or peptidomics approaches. However, the prediction of properties, such as those related to the bioactivity of these biomolecules enables either simplifying assumptions to be made or the application of new approaches for predicting bioactivity [13]. Here, we conducted an in silico prediction of physicochemical properties (hydrophobicity, hydrophilicity, intestinal stability, antiangiogenic, antihypertensive, and anti-inflammatory, using molecular dynamics simulations (MDS) and ensemble docking–virtual screening (VS) methods, with potential bioactive peptides from chia seed, and correlated these properties with human molecular (protein) targets (angiotensin converting enzyme (ACE), vascular endothelial growth factor (VEGF), glucocorticoid (GLUC), and mineralocorticoid (MINC) receptors). 

CSP library was initially built to reflect differences in physicochemical properties of selected peptides, and subsequentially those properties were used as descriptors. Several studies have reinforced the use of descriptors, showing the effectiveness of predicting a potential pharmaco-kinetic behavior based on ligand virtual screening techniques [14,15]. However, very few examples of peptides have shown the expected effectiveness [15,16]. Overall, a total of 1954 peptides were identified from the <3 kDa CSP fraction obtained from controlled enzymatic proteolysis. After the list of peptides were screened using Peptide Ranker, a total of 83 peptides were predicted as being bioactive (49 database peptides and 34 de novo peptides) according to their score (>0.8 threshold) (Table 1). This web server allowed us to predict the likehood of all the peptides for being bioactive using a neural network based on amino acid sequences and specific structure feature analysis [17]. Interestingly, it is important to note that the shortlisted database-peptides had a length between 7 and 17 amino acid residues, while shortlisted de novo peptides had a length between 4 and 6 amino acid residues. Scientific literature reports that bioactive peptides tend to be between 2–20 amino acids in length [18], with cores (hydrophobic or hydrophilic) of 4–5 amino acids that guide the interaction within the binding sites. Furthermore, several in silico bioactivity predictors have incorporated “peptide sequence length” as a descriptor into their quantitative structure relationship models [19,20]. Our de novo peptides fullfilled the necessary length to be considered as bioactive by different in silico predictors. 

The descriptors of the whole library were used to filter out the peptides with PreAIP, AHTpin, and AntioAngioPred servers for the prediction of their anti-inflammatory, antihypertensive, and antiangiogenic potentials before the ensemble docking analyses. In this scenario, physicochemical properties such as hydrophilicity, hydropathy, hydrophobicity, intestinal absorption (Table 2), and potential bioactivities (e.g., anti-angiogenic, anti-inflammatory, and antihypertensive) were prioritized, expecting that, among the screened peptide library, peptides with potential activity towards previously described protein targets could be identified [21]. Overall, it was observed that 52 and 42 of the 83 selected peptides were predicted as anti-inflamatoy and antihypertensive, respectively. This suggests that most of the selected peptides contained amino acids and structural features that contributed to those bioactivities. 

Particularly, in relation to the prediction of antiangiogenic peptides, due to the software limitation for processing peptides with lengths of at least of 5 amino acid residues, the majority of de novo peptides (i.e., 27) were not subjected to such analysis because of their short amino acid length. From the rest, 29 peptides were predicted as antiangiogenic, while 27 were considered as non-angiogenic. The process of angiogenesis is a vital step towards the formation of malignant cancer tumors [22]. Antiangiogenic peptides could therefore be promising candidates in the treatment for cancer. Thus, most of the peptides showed potential antiangiogenic (anti-tumor), anti-inflammatory, and antihypertensive properties. Other studies have applied in silico analysis of select peptides obtained from different protein sources to identify potential bioactivies such as amaranth [23], milk protein [24], chicken breast [17], and kerfir milk [25], with promising results. 

Dimensional reduction in calculated physicochemical properties was evaluated using Principal Component Analysis (PCA), which displayed a clustering hypothesis against three potential bioactivies, e.g., antiangiogenic (AAP), anti-inflamatory (AIP), and anti-hypertensive (AHT). Surprinsigly, physicochemical properties such as hydrophilicity, hydrophobicity and hydropathicity, clustered together with the aforementioned bioactivites, respectively (Figure 1). However, due to the high complexity of the analyzed biomolecules (e.g., tautomeric states, rotable bonds and side-chain diversity), the variance coverage was limited to nearly 72% (PC1-PC2). Furthermore, in chemoinformatics approaches toward natural products, these methodologies gave insights into the relationship between the calculated properties and possible bioactivity [26]. Acording to PCA and clustering analysis, four different molecular targets were selected; those targets were related to hypertension (angiotensin converting enzyme), inflammation (glucocorticoid and mineralocorticoid receptors), and angiogenesis (vascular endothelial growth factor).

A hybrid all-atom Molecular Dynamics Simulations and ensemble docking-virtual screening workflow was applied to each target; 300 ns of Gaussian Accelerated Molecular Dynamics simulations were performed on each target, in order to recover a major conformational diversity of protein structure that stems into more robust molecular docking analysis [27,28]. Three diverse conformations of each target were used as initial docking coordinates for the peptide library, and the score (Kcal mol^−1^) was averaged with the Vinardo Scoring Function. As shown in Figure 2, the avegare scoring values of targets and peptide library complexes showed lower energy values for VEGF and ACE (~−3 Kcal mol^−1^ and −5 Kcal mol^−1^, respectively) and higher values for mineralocortidoid and glucocorticoid receptors (~3 Kcal mol^−1^ and 12 Kcal mol^−1^, respectively). This difference may arise from more constrained binding sites on mineralocorticoid and glucocorticoid receptors, which generally evolve around the structure of phenanthrene core ligands [21,29]. 

Figure 3 shows the binding modes and ligand interaction diagrams of selected peptides from library vs. selected molecular targets. In the case of ACE (Figure 3A,B) and VEGF (Figure 3C,D), these targets displayed superficial binding sites (interfacial regions), which are highly dependent on the conformation of receptor proteins. Furthermore, the resulting binding modes for VEGF and ACE were more stable (lower energy) and in good agreement with those previously described [30]. Additionally, for ACE, the lowest energy peptide (SRPWPIDY) is posed on the same binding site of the spike protein for the SARS-CoV-2 virus (Figure 3A), displaying electrostatic interactions with amino acid residues (D30 and E37); whereas basic residues K26, H34 and R393 mediate atractions to D7 and COO^-^ terminal group of the Y8 residue. Moreover, the indole moiety of peptides was estabilized by aliphatic sidechains of V93 and backbone of T92 and K94. ACE was able to bind peptides through a flexible loop comprising A386–R393, which displays amphiphatic propierties, since it includes hydrophobic sidechains (A386, A387, F390) and backbone and amide sidechain from N388.

The resulting peptide library for VEGF binding modes corresponds to the previously described binding sites of natural products and pharmaceutical drugs [22]. The proposed binding sites of VEGF are located on the C-terminal and comprise residues such as Y124, P125, Y130, and M179, which mainly impose short-range van der Waals (vdW) interactions with ligands; whereas D126, E129 and R167 are involved in long range electrostatic interactions. All the displayed interactions contribute to the recognition of de novo peptides that contain positively charged residues such as R and K, along with aliphatic residues L, A and F (Figure 3C,D). However, certain short length peptides, as de novo, resemble the binding modes of natural ligands of nuclear receptors; those peptides displayed lower energy scores due to hydrogen bond interactions with W31 and T210 of the GLUC receptor, and hydrophobic interactions L39, I132, and M212 (Figure 3E,F). MINC showed the same pattern of hydrogen bonding through W57 and K147; whereas hydrophobic interactions with aliphatic sidechains of peptides were stabilized with A47, L46, F103, M51, V54, and L83 (Figure 3G,H).

The aim of this study was to integrate a multi-target-directed ligand approach, where peptides with lower binding energies can be shared as ligands towards the four evaluated human target proteins. To achieve this goal, the 20 lowest-binding, free energy peptides for each target protein were plotted and those that rely on the intersection of a Venn diagram (five known-sequence peptides (NNVFYPF, FNIVFPG, SRPWPIDY, QLQRWFR, GSRFDWTR) and five de novo peptides (DFKF, DLRF, FKAF, FRSF, QFRF)) were selected as potential multi-target-directed ligands, and turned into candidates for a future synthesis and in vitro evaluation (Figure 4). Those peptides can be considered as multifunctional peptides, which may be preferred over single-activity peptides, as they can simultaneously trigger, modulate, or inhibit multiple physiological pathways. The subset of selected peptides contained five know-sequence peptides (already described in databases and mass spectra data) and five de novo peptides (non previously described), which demonstrate that the in silico methods may assist in the identification of possible ligand hits.

Finally, we applied an absorption, distribution, metabolism, and excresion (ADME) prediction to evaluate drug-likeness parameters on the subset of selected multi-target directed peptides. The pharmaco-kinetic profiles of the complete subset assessed 25 different properties within the acceptable range for 95% of known drugs [31]. Table 3 lists the criteria used for each descriptor, as well as the calculated parameters and chemical similarity to approved drugs.

As shown in Table 3, the subset of selected peptides displayed lower calculated logP o/w partition coefficients that can be associated with increased permeability across membranes, despite having high molecular weight (>500 Da) and several hydrogen bond interactions. Similarly, different in silico techniques have addressed that the prediction of skin permeability by complex molecules is difficult to achieve; however, those predictions are based on comparisons to experimental data or complex algorithms based on quantitative structure–activity relationships (QSAR) [32,33]. Additonally, the calculated coefficient for binding to serum proteins and skin permeability are in the range of 95% of approved drugs, suggesting a positive interaction with epidermal tissues. The above obervations confirm that chia seed peptides could contribute in the improvement of the skin health and can be used as potential functional ingredients for development of cosmeceutical skin products, as previously demonstrated in vitro [9]. An interesting finding was that the entire subset of multi-target peptides showed structural similarity to antihypertensive and antiangiogenic drugs, such as lymecycline, everolimus and lisinopril, among others.

## 3. Materials and Methods

### 3.1. Materials

Alcalase (E.C. 3.4.21.62) and flavourzyme (E.C. 232-752-2) were acquired from Sigma Aldrich (St. Luis, MO, USA). Ultrafiltration units were purchased from Millipore (Bedford, MA, USA). Chia (*Salvia hispanica* L.) seeds were obtained from Healthworks (pesticide-free, Scottsdale, AZ, USA). All chemical reagents used in this study were analytical grade or HPLC grade.

### 3.2. Preparation of Chia Seed Peptides

Peptides from chia seed protein were obtained as described in detail by Urbizo-Reyes, San Martin-González, Garcia-Bravo, López Malo Vigil and Liceaga [7]. Briefly, chia seeds were devoid of mucilage using a combined treatment of ultrasound and vacuum-assisted filtration. Afterwards, chia seeds were defatted by a mechanical oil press and the resulting meal was proteolyzed using sequential enzymatic (alcalase followed by flavourzyme) microwave-assisted hydrolysis. The subsequent protein hydrolysate had a yield of 78.2% extracted protein. Resulting chia seed peptides (CSP) were freeze-dried and stored at −20 °C until used.

### 3.3. Peptidomics Analysis of Low Molecular Weight Peptides by Liquid Chromatography-Mass Spectrometry

Following proteolysis, low molecular weight peptides were obtained by ultrafiltration using centrifugal filters of 3 kDa cut-off membrane (regenerated cellulose, Amicon^®^ Ultra-4). Peptides present in the <3 kDa fraction were identified by liquid chromatography-mass spectrometry (LC-MS/MS) in the Proteomics Core Facility at the Indiana University School of Medicine (Indianapolis, Indiana, USA). Briefly, lyophilized samples of the <3 kDa fraction were resuspended in 0.1% of formic acid and reverse phase LC-MS was performed on the QE-Plus mass spectrometer. Peptide sequences were obtained from mass spectrometry data (raw files) using a database of all *Salvia* spp. proteins in the UniProt database and were also searched for de novo sequencing of peptides [8,9]. A total of 1954 peptides were identified (i.e., 1565 peptides using UniProt database and 389 de novo peptides).

### 3.4. In Silico Analysis of Identified Peptides for their Potential Bioactivities

First, each identified peptide was screened for their bioactive likelihood using the PeptideRanker bioinformatic tool (http://distilldeep.ucd.ie/PeptideRanker/, accessed on 6 March 2022), which estimates a bioactivity potential score based on a novel N-to-1 neural network. This server ranked all assessed peptides by giving scores ranging from 0 to 1. Values near 1 indicate peptides with high likelihood to be bioactive. Therefore, if the probabilities are ≥0.8, the peptide is said to be predicted as bioactive [34]. 

Selected peptides were subjected to in silico prediction and physicochemical characterization through different online servers. The predictions were made using the tools PreAIP (http://kurata14.bio.kyutech.ac.jp/PreAIP/, accessed on 7 March 2022) for their anti-inflammatory potential, AHTPIN (http://crdd.osdd.net/raghava/ahtpin/, accessed on 7 March 2022) for their antihypertensive potential, AntiAngioPred (http://webs.iiitd.edu.in/raghava/antiangiopred/, accessed on 8 March 2022) for their anti-angiogenic potential, HLP (http://crdd.osdd.net/raghava/hlp/, accessed on 8 March 2022) for their intestinal stability, and PlifePred (https://webs.iiitd.edu.in/raghava/plifepred/, accessed on 9 March 2022) for their plasma stability. Additionally, some physicochemical properties (i.e., hydrophobicity, hydropathicity, hydrophilicity, charge, and molecular weight) were determined using the AHTPIN tool. 

### 3.5. Construction of Human Molecular Targets and Identified Peptides 

For this analysis, the peptides identified in the previous step and some human targets were constructed. The three-dimensional structures of identified peptides sequences were modeled using the OPLS-AA forcefield with an alpha helix template (Maestro Suite v 2.12). The resulting structures were minimized using a conjugated gradient and Steepest Descent algorithms with the biomolecular simulation program, Amber20 package (https://ambermd.org/, accessed on 3 March 2022). The target crystallographic structures were Human Angiotensin Converting Enzymes, ACE-1 (PDB code: 1O8A), Vascular Endothelial Growth Factor, VEGF (PDB code: 2VPF), and Glucocorticoid and Mineralocorticoid Nuclear Receptors (PDB code: 1P93 and 2AA2, respectively). Modeling of missing or incomplete residues and disulfide bonds were carried out with Maestro Suite v 2.12, (Schrödinger, LLC, New York, NY, USA) using OPLS-AA force field to guide the structure completion. The tritiable residue protonation was calculated using the PROPKA web-server at pH 7 (e.g., histidine was protonated on the delta/epsilon nitrogen) depending on the polar environment. 

All targets were prepared in octahedral water boxes 15 Å buffered with TIP3P explicit water molecules and neutralized with Na+ or Cl- ions using Tleap routine from AmberTools 20. The systems were neutralized with a salt solution of 0.15 M NaCl. The resulting systems comprised between 100,000–172,000 atoms. 

### 3.6. Molecular Dynamics and Ensemble Docking Calculations

All Molecular Dynamics calculations were carried out in Amber20 using the amberff14 and GAFF force fields with PMEMD.cuda software. Initial MD stages (minimization and thermalization) were performed at 1 atm of pressure and 298 °K. The minimization consisted of 5000 steps using the steepest-descendent method, followed by 10,000 with the conjugate gradient method. Covalent bonds between hydrogen and heavy atoms were constrained using the SHAKE algorithm. Initial simulations steps applied an NVT ensemble, coupled with a Langevin thermostat, to reach and maintain a constant temperature (298 °K). Linear increases in the temperature from 0 to 298 °K, in intervals of 50 °K, were applied along 5,000,000 steps. The atoms were restricted in protein alpha carbons atoms with 5 Kcal/mol/A3 and 2.5 Kcal/mol/A3. An isothermal-semi isobaric ensemble (NPT) simulation, using the Berendsen barostat, was coupled to maintain 1 atm of pressure keeping restrains. Afterwards, restrains were reduced by 2 Kcal/mol/A3 units for proteins 250,000 steps until restriction disappeared. The systems continued for 10 ns more in simulation in the anisotropic ensemble at 1 atm, keeping track of Cα root-mean-square deviation (RMSD), potential energy and system density as markers of stable behavior. Long-range electrostatic interactions were calculated, periodic boundary conditions, and particle mesh Ewald (PME) methods using a 12 Å cutoff and force switch on a 10 Å radius. Time steps of 2.0 fs were set for production simulations. Production calculations were performed along 300 ns for each target using Gaussian Accelerated Molecular Dynamics (GaMD) to enhance conformational sampling. After each GaMD production, the carbon alpha root mean square deviation (A) was calculated and projected onto first two principal components using Bio3D package for selection of three representative conformations for ensemble docking.

Binding modes and score calculations (Autodock Vina scoring function) were performed with Smina v 1.0 package for Ensemble docking—Virtual screening. All representative conformation for each previously selected target was used for docking calculations. The search parameters were 0.375 Å mesh step, 64 as exhaustiveness value, and a maximum value of 25,000,000 evaluations. The octahedral search space dimensions for all targets were set to 25 × 25 × 40 Å centered on each identified active site or interaction regions, as described for ACE enzymes. All scores were approximated using consensus scoring approximation.

## 4. Conclusions

In this work, we presented a pipeline based on in silico analysis upon a set of chia seed peptides identified using a peptidomics and bioinformatics approach that sought to screen potential bioactivities and correlate them with pharmacological (drug) targets. Structural analysis and physicochemical prediction of descriptors allowed us to select probable molecular targets and virtually screen their intermolecular interactions through molecular dynamics simulations and ensemble docking. With this approach, we were able to identify 10 multifunctional chia seed peptides (NNVFYPF, FNIVFPG, SRPWPIDY, QLQRWFR, GSRFDWTR, DFKF, DLRF, FKAF, FRSF, QFRF) with low binding free energy (ca. −5 Kcal mol^−^^1^) and stable intermolecular interactions during formation of a ligand-receptor complex. Furthermore, this information corroborates the high in vitro biological activity previously reported for chia seed peptides and foresees sequences found in the peptide fractions that can provide multifunctional bioactivities towards human protein receptors involved in chronic diseases such as hypertension and inflammation. However, further research will be required to test the specified peptide sequences using in vivo models. Nevertheless, this study provides promising use of bioinformatics and in silico analyses to establish ligand-receptor interactions from new functional ingredients and emerging protein sources.

## Figures and Tables

**Figure 1 ijms-23-07288-f001:**
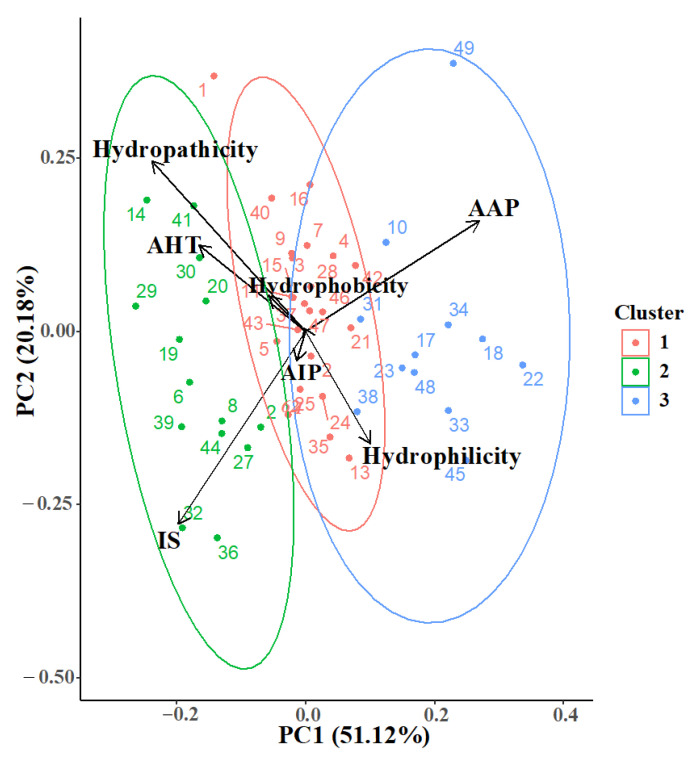
Principal Component Analysis of calculated physicochemical and bioactivity properties. AAP, antiangiogenic; AHT, antihypertensive; AIP, anti-inflammatory; IS, Intestinal Stability. Each dot corresponds to a peptide in the library (Table 1).

**Figure 2 ijms-23-07288-f002:**
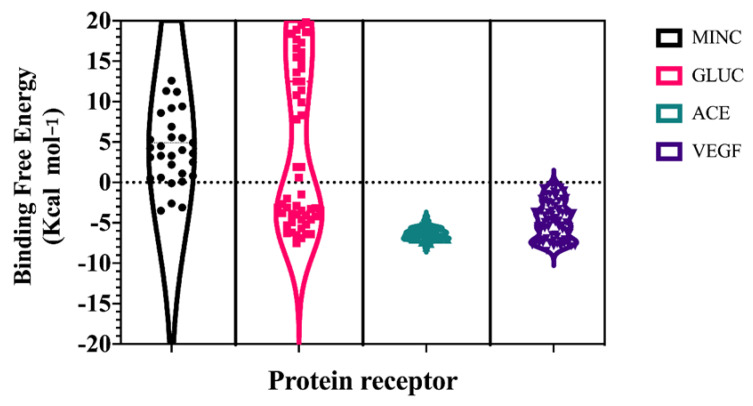
Ensemble docking-virtual screening scores of chia seed peptides towards select human protein targets: MINC (mineralocorticoid), GLUC (glucocorticoid), ACE (angiotensin converting enzyme), and VEGF (vascular endothelial growth factor).

**Figure 3 ijms-23-07288-f003:**
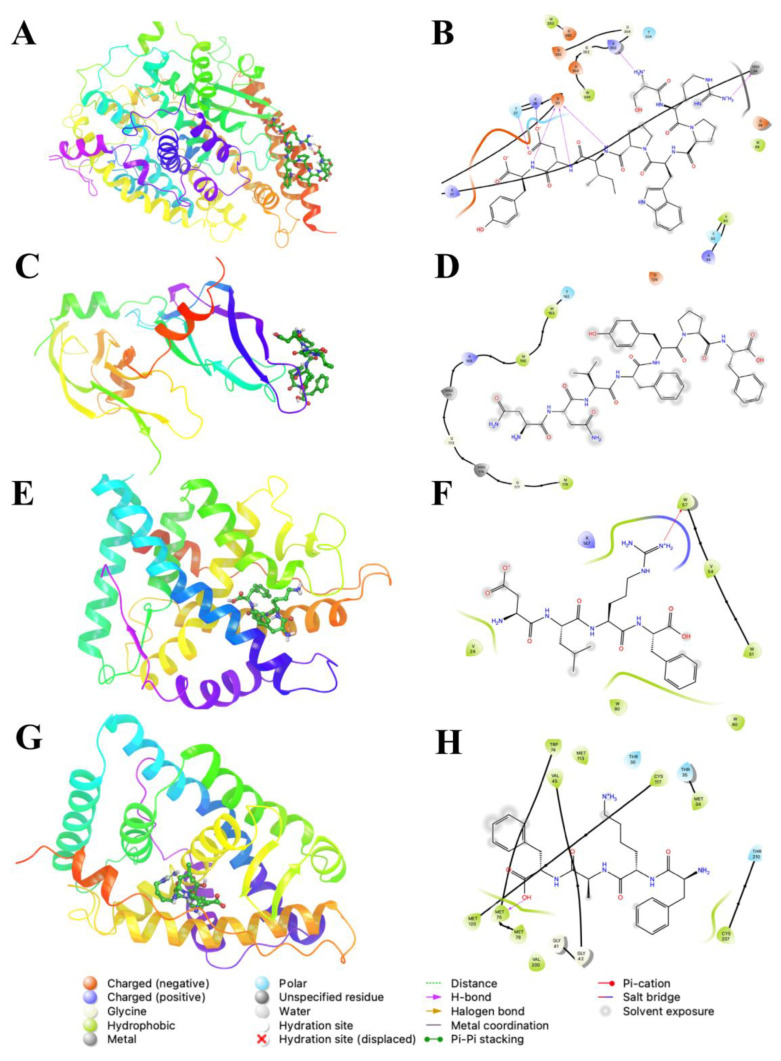
Binding modes (left) and ligand interaction diagrams (right) of selected peptides from library vs. selected molecular targets. (**A**,**B**): Angiotensin converting enzyme (ACE) in complex with WPRSPIDY; (**C**,**D**): Vascular Endothelial growth factor (VEGF) in complex with FVNYPF; (**E**,**F**): Glucocorticoid receptor (GLUC) in complex wit KFAF; (**G**,**H**): Mineralocorticoid receptor (MINC) in complex with RLDF. Ligand interactions were calculated around a 5 Å distance cutoff.

**Figure 4 ijms-23-07288-f004:**
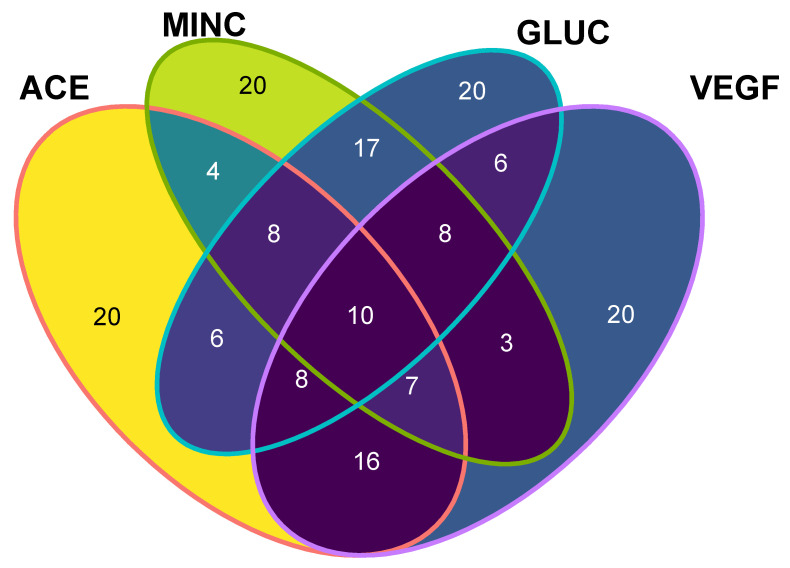
Venn diagram of identified peptides with their human target protein interactions. The numbers indicate independent subsets of shared peptides with the lowest binding free energy among molecular targets. Ten different peptides comprise the subset of multi-target directed ligands. Angiotensin converting enzyme (ACE), Mineralocorticoid receptor (MINC), Glucocorticoid receptor (GLUC), Vascular Endothelial growth factor (VEGF).

**Table 1 ijms-23-07288-t001:** In silico analysis results of selected peptide sequences from chia seed.

No.	Peptide Sequence	Peptide Ranker Score	PreAIP		AntiAngio-Pred		AHTpin	
			Score	Prediction	Score	Prediction	Score	Prediction
**Database peptides**								
1	FNLVFFLL	0.951	0.416	AIP	−0.21	Non-AAP	−0.58	Non-AHT
2	EGDVFWIPRF	0.940	0.418	AIP	−0.5	Non-AAP	−0.12	Non-AHT
3	DHFPFIY	0.933	0.518	AIP	0.04	AAP	0.47	AHT
4	EGGIWPF	0.929	0.344	AIP	0.48	AAP	−0.1	Non-AHT
5	GFEWITF	0.922	0.57	AIP	0.47	AAP	−0.89	Non-AHT
6	GLDFPELPLGM	0.919	0.481	AIP	−0.61	Non-AAP	1.31	AHT
7	GQTPLFPRIF	0.912	0.412	AIP	0.41	AAP	0.65	AHT
8	GDAHYDPLFPF	0.909	0.323	Non-AIP	−1.23	Non-AAP	0.95	AHT
9	NNVFYPF	0.903	0.344	AIP	−0.34	Non-AAP	0.22	AHT
10	EYPPLGRF	0.901	0.395	AIP	1.01	AAP	1.04	AHT
11	KPLPFELF	0.898	0.409	AIP	0.6	AAP	0.55	AHT
12	DVWDPFQDFPL	0.895	0.461	AIP	0.11	AAP	0.41	AHT
13	SDKNGYFF	0.883	0.418	AIP	−0.82	Non-AAP	−1.16	Non-AHT
14	VPIPVPLPF	0.883	0.318	Non-AIP	−0.24	Non-AAP	2.29	AHT
15	SNVFDPF	0.876	0.338	Non-AIP	−0.87	Non-AAP	−0.06	Non-AHT
16	TPLFPRIF	0.876	0.393	AIP	1.03	AAP	0.58	AHT
17	DQNPRSFFL	0.873	0.444	AIP	1	AAP	−0.67	Non-AHT
18	QLQRWFR	0.871	0.519	AIP	2.22	AAP	−0.62	Non-AHT
19	GFEWVAF	0.868	0.59	AIP	−0.94	Non-AAP	0.3	AHT
20	SFNLPIL	0.867	0.408	AIP	−0.4	Non-AAP	−0.15	Non-AHT
21	QEGGIWPF	0.863	0.37	AIP	0.39	AAP	−0.52	Non-AHT
22	GSRFDWTR	0.858	0.488	AIP	2.17	AAP	−1.36	Non-AHT
23	ADFYNPR	0.853	0.303	Non-AIP	0.92	AAP	0.44	AHT
24	APSKDAPMF	0.851	0.452	AIP	−0.16	Non-AAP	−0.24	Non-AHT
25	GFEWITFK	0.847	0.578	AIP	0.28	AAP	−0.66	Non-AHT
26	NGFEWITF	0.842	0.549	AIP	0.4	AAP	−1.03	Non-AHT
27	VNEGDVFWIPRF	0.841	0.414	AIP	−1.1	Non-AAP	−0.52	Non-AHT
28	SSNVFDPF	0.841	0.304	Non-AIP	−0.93	Non-AAP	−0.17	Non-AHT
29	FNIVFPG	0.839	0.385	AIP	−1.68	Non-AAP	0.76	AHT
30	VPVFPPPLN	0.837	0.435	Non-AIP	−0.26	Non-AAP	2	AHT
31	GIDIPPPR	0.835	0.316	Non-AIP	0.55	AAP	0.47	AHT
32	APAEKGFAGF	0.832	0.402	AIP	−1.39	Non-AAP	0.23	AHT
33	DQNPRSFF	0.830	0.433	AIP	1.05	AAP	−0.92	Non-AHT
34	SRPWPIDY	0.827	0.486	AIP	2.26	AAP	−0.04	Non-AHT
35	QNGFEWITF	0.825	0.573	AIP	0.46	AAP	−1.47	Non-AHT
36	RPGDVFVFPR	0.825	0.383	AIP	−0.98	Non-AAP	0.22	AHT
37	DNGIIYPW	0.823	0.32	Non-AIP	−0.36	Non-AAP	0.15	AHT
38	NPQAGRF	0.822	0.376	AIP	−0.43	Non-AAP	0.03	AHT
39	APVGSPVGSTGGNFGVF	0.817	0.476	AIP	−1.1	Non-AAP	0.39	AHT
40	APPPVLAL	0.816	0.396	AIP	0.61	AAP	0.07	AHT
41	FPLLNYL	0.813	0.554	AIP	−0.18	Non-AAP	1.31	AHT
42	RNNVFYPF	0.811	0.378	AIP	0.61	AAP	0.49	AHT
43	GNIFRGL	0.811	0.452	AIP	−0.34	Non-AAP	−0.6	Non-AHT
44	FPGLADRM	0.810	0.333	Non-AIP	−1.01	Non-AAP	0.67	AHT
45	SNEWDPSFR	0.806	0.393	AIP	0.81	AAP	−1.34	Non-AHT
46	SMLSPHW	0.806	0.42	AIP	0.22	AAP	−0.24	Non-AHT
47	SLDVWDPFQDFPL	0.804	0.464	AIP	0.44	AAP	0.59	AHT
48	SPDLIRRM	0.803	0.399	AIP	1.12	AAP	−0.74	Non-AHT
49	FGNVFKGM	0.803	0.32	Non-AIP	−1.68	Non-AAP	−0.8	Non-AHT
**De novo peptides**								
1	QFRF	0.980	0.361	AIP	ND	-	−0.05	Non-AHT
2	FDRF	0.978	0.301	Non-AIP	ND	-	−0.79	Non-AHT
3	GRPW	0.971	0.266	Non-AIP	ND	-	0.84	AHT
4	FWDR	0.964	0.347	AIP	ND	-	−0.74	Non-AHT
5	FRSF	0.962	0.339	Non-AIP	ND	-	−0.76	Non-AHT
6	GPHW	0.958	0.354	AIP	ND	-	1	AHT
7	KPPF	0.955	0.294	Non-AIP	ND	-	2.09	AHT
8	WLPR	0.943	0.306	Non-AIP	ND	-	1.13	AHT
9	FWDH	0.938	0.318	Non-AIP	ND	-	−0.61	Non-AHT
10	FDKF	0.937	0.32	Non-AIP	ND	-	−0.87	Non-AHT
11	FRGL	0.937	0.318	Non-AIP	ND	-	−0.38	Non-AHT
12	DFKF	0.932	0.336	Non-AIP	ND	-	−0.87	Non-AHT
13	KDFLFP	0.929	0.352	AIP	−0.03	Non-AAP	−0.79	Non-AHT
14	EFRF	0.922	0.289	Non-AIP	ND	-	0.99	AHT
15	APHW	0.918	0.389	AIP	ND	-	0.03	AHT
16	RPAF	0.909	0.28	Non-AIP	ND	-	0.89	AHT
17	ARGW	0.908	0.317	Non-AIP	ND	-	−0.76	Non-AHT
18	FKAF	0.907	0.31	Non-AIP	ND	-	−0.67	Non-AHT
19	WEFLTF	0.907	0.329	Non-AIP	0.43	AAP	−0.89	Non-AHT
20	HVFF	0.878	0.349	AIP	ND	-	−0.21	Non-AHT
21	WAPH	0.873	0.313	Non-AIP	ND	-	0.99	AHT
22	RPSF	0.872	0.334	Non-AIP	ND	-	0.7	AHT
23	HPAYW	0.871	0.392	AIP	0.1	AAP	1.81	AHT
24	DLRF	0.863	0.319	Non-AIP	ND	-	−0.56	Non-AHT
25	QLRF	0.863	0.344	AIP	ND	-	0.21	AHT
26	GKFL	0.850	0.316	Non-AIP	ND	-	−0.48	Non-AHT
27	QRYF	0.848	0.327	Non-AIP	ND	-	0.89	AHT
28	FWDNH	0.834	0.345	AIP	−0.63	Non-AAP	0.05	AHT
29	FPLK	0.834	0.336	Non-AIP	ND	-	1.07	AHT
30	RAFL	0.831	0.349	AIP	ND	-	−0.42	Non-AHT
31	FPLLN	0.820	0.483	AIP	−0.1	Non-AAP	0.22	AHT
32	WDPSYR	0.816	0.325	Non-AIP	2.67	AAP	0.11	AHT
33	GLKF	0.810	0.348	AIP	ND	-	−0.48	Non-AHT
34	HPNPRL	0.808	0.358	AIP	0.51	AAP	0.98	AHT

ND: Not determined due to server limitations; PreAIP: Peptides with a score > 0.342 are predicted as anti-inflammatory peptides (AIP); AngioPred: a positive score is predicted as an antiangiogenic peptide (AAP); AHTpin: a positive score is predicted as an antihypertensive peptide (AHT); a score > 1 is predicted as high stability, while a score < 1 is predicted as normal stability.

**Table 2 ijms-23-07288-t002:** Physicochemical chracteristics of selected chia seed peptide sequences.

No.	Peptide Sequence	Intestinal Stability	Hydrophobicity	Hydropathicity	Hydrophilicity	Charge	Molecular Weight
**Database peptides**							
1	FNLVFFLL	0.73	0.42	2.56	−1.78	0	1012.35
2	EGDVFWIPRF	2.391	−0.02	−0.01	−0.27	−1	1265.57
3	DHFPFIY	1.094	0.11	0.07	−0.94	−1	938.13
4	EGGIWPF	0.945	0.19	0.07	−0.67	−1	805
5	GFEWITF	2.512	0.24	0.66	−1.09	−1	899.11
6	GLDFPELPLGM	2.79	0.12	0.46	−0.29	−2	1188.54
7	GQTPLFPRIF	1.079	−0.01	0.16	−0.58	1	1175.52
8	GDAHYDPLFPF	2.144	0.02	−0.35	−0.37	−2	1278.52
9	NNVFYPF	0.798	0.06	−0.01	−1.2	0	900.09
10	EYPPLGRF	0.392	−0.15	−0.79	−0.07	0	978.21
11	KPLPFELF	1.859	0.05	0.32	−0.33	0	990.3
12	DVWDPFQDFPL	1.608	−0.03	−0.41	−0.23	−3	1378.65
13	SDKNGYFF	1.183	−0.17	−0.98	−0.1	0	977.14
14	VPIPVPLPF	2.283	0.3	1.46	−1.01	0	978.35
15	SNVFDPF	0.585	0.01	0.06	−0.43	−1	824.97
16	TPLFPRIF	1.143	0.05	0.69	−0.75	1	990.3
17	DQNPRSFFL	1.023	−0.27	−0.89	−0.01	0	1123.34
18	QLQRWFR	1.066	−0.48	−1.47	−0.19	2	1033.3
19	GFEWVAF	2.466	0.27	0.97	−1.06	−1	855.06
20	SFNLPIL	2.342	0.2	1.29	−1.06	0	803.04
21	QEGGIWPF	1.213	0.08	−0.38	−0.56	−1	933.15
22	GSRFDWTR	0.667	−0.44	−1.56	0.38	1	1024.21
23	ADFYNPR	1.132	−0.33	−1.4	0.13	0	882.02
24	APSKDAPMF	1.538	−0.09	−0.34	0.17	0	963.22
25	GFEWITFK	2.682	0.07	0.09	−0.58	0	1027.3
26	NGFEWITF	2.499	0.13	0.14	−0.93	−1	1013.23
27	VNEGDVFWIPRF	2.24	−0.02	0.05	−0.33	−1	1478.84
28	SSNVFDPF	0.2	−0.02	−0.05	−0.34	−1	912.06
29	FNIVFPG	2.136	0.28	1.26	−1.16	0	793.02
30	VPVFPPPLN	1.974	0.14	0.57	−0.79	0	979.3
31	GIDIPPPR	0.964	−0.13	−0.53	0.3	0	864.1
32	APAEKGFAGF	3.43	0.05	0.12	−0.05	0	994.24
33	DQNPRSFF	1.044	−0.36	−1.48	0.21	0	1010.17
34	SRPWPIDY	1.413	−0.22	−1.21	−0.15	0	1033.25
35	QNGFEWITF	2.539	0.04	−0.27	−0.8	−1	1141.38
36	RPGDVFVFPR	3.379	−0.19	−0.21	0.1	1	1189.51
37	DNGIIYPW	0.959	0.07	−0.28	−0.76	−1	977.19
38	NPQAGRF	0.929	−0.31	−1.27	0.06	1	788.95
39	APVGSPVGSTGGNFGVF	2.935	0.14	0.53	−0.56	0	1549.95
40	APPPVLAL	1.457	0.24	1.32	−0.76	0	777.06
41	FPLLNYL	1.841	0.22	1.11	−1.43	0	879.14
42	RNNVFYPF	0.769	−0.17	−0.58	−0.67	1	1056.29
43	GNIFRGL	1.294	−0.03	0.33	−0.41	1	775.99
44	FPGLADRM	2.43	−0.09	0.04	−0.01	0	906.16
45	SNEWDPSFR	0.992	−0.37	−1.81	0.43	−1	1137.29
46	SMLSPHW	1.339	0.02	−0.23	−0.91	0	857.09
47	SLDVWDPFQDFPL	1.558	0	−0.12	−0.31	−3	1578.91
48	SPDLIRRM	1.23	−0.38	−0.59	0.55	1	987.27
49	FGNVFKGM	1.854	0.07	0.44	−0.57	1	899.19
**De novo peptides**							
1	QFRF	ND	−0.31	−0.6	−0.45	1	596.73
2	FDRF	ND	−0.32	−0.6	0.25	0	583.68
3	GRPW	ND	−0.33	−1.85	−0.1	1	514.64
4	FWDR	ND	−0.38	−1.52	0.02	0	622.73
5	FRSF	ND	−0.2	0.07	−0.42	1	555.67
6	GPHW	ND	0.01	−1.53	−0.97	0	495.6
7	KPPF	ND	−0.16	−1.07	0.12	1	487.64
8	WLPR	ND	−0.23	−0.8	−0.55	1	570.74
9	FWDH	ND	−0.04	−1.2	−0.85	−1	603.69
10	FDKF	ND	−0.15	−0.45	0.25	0	555.67
11	FRGL	ND	−0.11	0.42	−0.33	1	491.63
12	DFKF	ND	−0.15	−0.45	0.25	0	555.67
13	KDFLFP	1.208	−0.29	−0.6	0.25	0	597.71
14	EFRF	ND	0.04	−0.97	−1.1	0	509.62
15	APHW	ND	−0.02	0.07	−0.13	0	765.97
16	RPAF	ND	−0.24	−0.38	0	1	489.61
17	ARGW	ND	−0.25	−1	−0.22	1	488.6
18	FKAF	ND	0.09	0.88	−0.62	1	511.66
19	WEFLTF	1.16	0.22	0.72	−1.27	−1	842.04
20	HVFF	ND	0.34	1.65	−1.75	0	548.69
21	WAPH	ND	0.04	−0.98	−1.1	0	509.62
22	RPSF	ND	−0.37	−1.02	0.2	1	505.61
23	HPAYW	1.398	0.03	−1.04	−1.34	0	672.81
24	DLRF	ND	−0.33	−0.35	0.43	0	549.66
25	QLRF	ND	−0.33	−0.35	−0.28	1	562.71
26	GKFL	ND	0.05	0.57	−0.33	1	463.62
27	QRYF	ND	−0.46	−1.63	−0.4	1	612.73
28	FWDNH	1.425	−0.16	−1.66	−0.64	−1	717.81
29	FPLK	ND	−0.01	0.28	−0.32	1	503.68
30	RAFL	ND	−0.09	0.97	−0.45	1	505.65
31	FPLLN	1.991	0.19	1.06	−1.18	0	602.78
32	WDPSYR	1.486	−0.4	−2.1	0.1	0	822.95
33	GLKF	ND	0.05	0.57	−0.33	1	463.62
34	HPNPRL	1.534	−0.4	−1.77	0.15	1	732.91

ND: Not determined due to server limitations (peptides ≤ 4 amino acid residues).

**Table 3 ijms-23-07288-t003:** Calculated drug-likeness parameter for subset of 10 selected peptide sequences.

Parameters	NNVFYPF	FNIVFPG	SRPWPIDY	QLQRWFR	GSRFDWTR	DFKF	DLRF	FKAF	FRSF	QFRF
*M*_W_ (g/mol)	899.9	792.93	899.07	10.33.19	1024.1	555.63	549.62	511.62	555.63	596.68
H-bond donors	8	6	8.5	17.5	14.5	5.75	7.75	5.75	7.75	9.75
H-bond acceptors	19.25	17	19	22.5	22.4	10.25	11.25	9.25	10.95	12.75
logP o/w ^a^	−3.56	−1.83	−2.79	−5.68	−4.64	−1.06	−1.83	−1.43	−1.28	−3.14
logS wat ^b^	−1.87	−3.77	−0.49	−1.18	−0.24	−0.68	−0.64	−0.79	−2.72	−0.96
NlogK has Serum Protein Binding ^c^	−2.39	−1.88	−2.53	−3.35	−3.18	−1.23	−1.54	−1.02	−1.26	-1.75
Apparent Caco-2 Permeability (nm/s) ^d^	0	0	0	0	0	0	0	0	0	0
Apparent MDCK Permeability (nm/s) ^e^	0	0	0	0	0	0	0	0	0	0
logK*p* for skin permeability ^f^	−9.04	−6.92	−8.53	−14.29	−13.53	−8.69	−9.95	−8.26	−8.62	−10.60
Qualitative Model for Human Oral Absorption	Low	Low	Low	Low	Low	Low	Low	Low	Low	Low
Most similar pharmaceutical drugs	Troxerutin, Voglibose, Monoxerutin	Lymecycline, Troxerutin, Proglumetacin	Razoxane, Hexoprenaline, Dihydralazine	Everolimus, Amiodarone, Fenethylline	Everolimus, Droxidopa, Polaprezinc	Hexoprenaline, Lisinopril, Lymecycline	Hexoprenaline, Voglibose, Lymecycline	Hexoprenaline, Lisinopril, Lymecycline	Aminopterin, Lymecycline, Hexobendine	Hexobendine, Hexoprenaline, Lymecycline

^a^ Predicted logarithm of partitioning coefficient for octanol/water phases (range for 95% of drugs: −2.0 to 6.0). ^b^ Predicted logarithm of aqueous solubility in mol/dm^3^ (range for 95% of drugs: −6.0 to 0.5). ^c^ Predicted logarithm of serum protein binding (range for 95% of drugs: −1.5 to 1.5). ^d^ Predicted apparent Caco-2 cell rate permeability in nm/s (range for 95% of drugs: <25, >500). ^e^ Predicted apparent MDCK cells rate permeability in nm/s (range for 95% of drugs: <25, >500). ^f^ Predicted apparent for skin permeability rate permeability Kp in cm/h).

## Data Availability

Not applicable.

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
