# Peer review of "Multifunctional Analysis of Chia Seed (Salvia hispanica L.) Bioactive Peptides Using Peptidomics and Molecular Dynamics Simulations Approaches"

_ijms, 2022, doi:10.3390/ijms23137288_

Round 1

Reviewer 1 Report

This is good basic research to understand the protein structure and function,  consider:

revise objective statement line 57-62.

table 2, provide a note for ND?

conclusion is weak and needs improvement…

Reviewer 2 Report

Dear Authors

I appreciated the huge work of characterization done on these peptides as primary sequences and the in silico work. The language needs revision for grammar mistakes and pre sample preparation should report the extraction yields for proteic mixtures. Since contaminants can be present in organic material, can the authors perform a protein tritration?.

Overall the manuscript is interesting and deserves consideration after minor changes.

Reviewer 3 Report

The manuscript presents a study by which following proteolysis, low molecular weight peptides were obtained from Chia seeds. 1,954 peptide sequences were identified from mass spectrometry data (i.e., 1,565 peptides using UniProt database and 389 de novo peptides). The work focuses mainly on in silico methodologies (i.e., physicochemical properties prediction, molecular dynamics and molecular docking simulations) to predict potential bioactivities associated to chia seed peptides. As result,10 chia seed peptides were proposed as potential multifunctional bioactivities towards human protein receptors involved in chronic diseases. This study is interesting and may be of interest to the scientific community. However, certain completions or corrections of the manuscript are required as follows:

Figure 1 should appear after it has been mentioned in the text (row 150).

Rows 175-176; 199 - 202: “Figure 3 shows the binding modes and ligand interaction diagrams of the peptide library vs. selected molecular targets” Figure 3 (BDFH) shows only 4 peptides, please specify their name.

Row 197: It was explicitly discussed about ACE and VEGF binding modes, a brief discussion about GLUC and MINC (Figure 3) would also be necessary.

Row 239: What does the a15 and a16 mean?

Row 313- 314: “Vascular Endothelial Growth Factor, VEGF (PDB code: 6CM4)” The PDB code: 6CM4 belongs to the D2 Dopamine Receptor, not to Vascular Endothelial Growth Factor, VEGF. Please add the correct PDB code for the VEGF.
